

# Effectiveness of in-group *versus* individually administered pain neuroscience education on clinical and psychosocial outcomes in patients with chronic low back pain: randomized controlled study protocol

Joaquín Salazar-Méndez[1,2], Iván Cuyul-Vásquez[3,4], Felipe Ponce-Fuentes[5], Rodrigo Núñez-Cortés[6], Guillermo Mendez-Rebolledo[1,2] and Jorge Fuentes[7,8]

[1] Laboratorio de Investigación Somatosensorial y Motora, Escuela de Kinesiología, Facultad de Salud, Universidad Santo Tomás, Talca, Chile
[2] Escuela de Kinesiología, Facultad de Salud, Universidad Santo Tomás, Talca, Chile
[3] Departamento de Procesos Terapéuticos, Facultad de Ciencias de la Salud, Universidad Católica de Temuco, Temuco, Chile
[4] Facultad de Ciencias de la Salud, Universidad Autónoma de Chile, Temuco, Chile
[5] Facultad de Ciencias, Escuela de Kinesiología, Universidad Mayor, Temuco, Chile
[6] Departament of Physical Therapy, Faculty of Medicine, University of Chile, Santiago, Chile
[7] Clinical Research Lab, Department of Physical Therapy, Catholic University of Maule, Talca, Chile
[8] Faculty of Rehab Medicine, University of Alberta, Edmonton, Canada

Corresponding author
Guillermo Mendez-Rebolledo,
guillermomendezre@santotomas.cl

## ABSTRACT

**Objective**. (1) This trial will compare the clinical and psychosocial effectiveness of in-group and individually pain neuroscience education (PNE) in patients with chronic low back pain (CLBP). In addition, (2) the influence of social determinants of health on post-treatment results will be analyzed.

**Methods**. A three-arm randomized controlled trial will be conducted. Sixty-nine participants with CLBP will be recruited in a 1:1:1 ratio. Participants, assessor, and statistician will be blinded to group assignment. The PNE intervention will be adapted to the context of the participants. An experimental group ($n = 33$) will receive PNE in an in-group modality, the other experimental group ($n = 33$) will receive PNE in an individually modality and the control group ($n = 33$) will continue with usual care. Additionally, participants will be encouraged to stay active by walking for 20–30 min 3–5 times per week and will be taught an exercise to improve transversus abdominis activation (bracing or abdominal following). The outcome measures will be fear avoidance and beliefs, pressure pain threshold, pain self-efficacy, catastrophizing, pain intensity, and treatment expectation. Outcome measures will be collected at one-week before intervention, immediately post-intervention, and four-weeks post-intervention.

**Conclusion**. The innovative approach of PNE oriented to fear beliefs proposed in this study could broaden the application strategies of this educational therapeutic modality. **Impact**. Contextualized PNE delivered by physical therapist could be essential to achieve

a good cost-effectiveness ratio of this intervention to improve the clinical condition of people with CLBP.

## INTRODUCTION

Chronic low back pain (CLBP) is the leading cause of disability worldwide (*Hoy et al., 2010*). It has been shown that approximately 67% of the general population have low back pain for more than three months (*Itz et al., 2013*). The disability in these individuals varies from 11% to 76% (*Côté et al., 2008*; *Wynne-Jones, Dunn & Main, 2008*), with a significant impact of psychosocial factors on their response to treatment (*Hill & Fritz, 2011*; *Alhowimel et al., 2018*). Furthermore, social determinants of health (SDH) are a determining factor on the symptomatology of CLBP (*Karran, Grant & Moseley, 2020*), associated with substantial social and healthcare expenses (*Luo et al., 2004*).

The psychological factors of the fear-avoidance model, such as catastrophizing, beliefs, fear of movement, and self-efficacy, are important determinants of symptom perception and disability. These psychological factors can impede the recovery process (*Pincus et al., 2002b*; *Pincus et al., 2002a*; *Leeuw et al., 2007*; *Zale & Ditre, 2015*), cause physical deconditioning, and perpetuate pain in people with CLBP (*Vlaeyen & Linton, 2000*). From this perspective, catastrophizing shows a moderate correlation with pain intensity and disability in patients with CLBP (*Meyer et al., 2009*). It is also a significant predictor of both pain intensity and disability (*Picavet, Vlaeyen & Schouten, 2002*). Furthermore, fear avoidance and self-efficacy beliefs predict disability and mediate the disability-pain intensity relationship (*Denison, Åsenlöf & Lindberg, 2004*; *Woby, Urmston & Watson, 2007*; *Costa et al., 2011*; *Lee et al., 2015*), making them potential targets for clinical interventions (*Pincus et al., 2002a*). In this sense, it has been shown that the management of avoidance beliefs has an effect in reducing disability and pain in patients with CLBP (*Wertli et al., 2014*), and high levels of self-efficacy may prevent the vicious cycle of deconditioning and pain perpetuation (*Woby et al., 2004*). Furthermore, it has been found that there are independent and interdependent relationships between SDH and CLBP mainly for educational level and socioeconomic level (*Karran, Grant & Moseley, 2020*).

In chronic pain rehabilitation, it is essential to address the factors mentioned above (*McCracken, 2005*). In this sense, the biomedical approach to education may promote the fear-avoidance model (*Louw et al., 2011*), while the pain neuroscience education (PNE) promotes patients' understanding of chronic pain and changes maladaptive thoughts and cognitions (*Moseley, 2002*; *Meeus et al., 2010*) with a biopsychosocial approach limiting the fear-avoidance model (*Louw et al., 2016b*). PNE has shown positive results on the kinesiophobia, catastrophizing, pain intensity, disability, and physical performance in patients with CLBP (*Moseley, Nicholas & Hodges, 2004*; *Ryan et al., 2010*; *Louw et al., 2011*; *Malfliet et al., 2017*; *Rufa, Beissner & Dolphin, 2018*; *Núñez Cortés et al., 2023a*; *Nuñez*

*Cortés et al., 2023b*). A recent meta-analysis has shown that the in-group PNE had better results than individual PNE for kinesiophobia (*Romm et al., 2021*), and it has been identified, in other types of health education, that there are greater benefits in those people who received the in-group intervention (*Riemsma, Taal & Rasker, 2003*), probably supported by the fact that educational sessions conducted in-group modality can facilitate learning through social observation of positive behaviors exhibited by other members within the group (*Romm et al., 2021*; *Salazar-Méndez et al., 2024*). However, there are no primary studies comparing the two PNE modalities directly. On the other hand, despite increasing evidence that SDH are influential factors in clinical health outcomes to a greater extent than the quality and availability of medical care (*Daniel, Bornstein & Kane, 2018*), there are no studies that analyze specifically the influence of SDH factors (*e.g.*, educational level) on the effectiveness of PNE (*Salazar-Méndez et al., 2024*).

We hypothesize that there will be significant differences in favor of the intervention in-group modality compared to the individual modality. Furthermore, the effects will be influenced by the social determinants of health in both experimental groups.

### Study objectives

The objectives of this trial are: (1) to compare the clinical and psychosocial effectiveness of in-group modality and individual modality of pain neuroscience education (PNE) in patients with CLBP; (2) To analyze the influence of social determinants of health on post-treatment results.

## METHODS

### Study design

This randomized controlled trial study protocol has a three-group comparison design, with a control group, individual intervention group, and in-group intervention group. It has been designed according to the Standard Protocol Items for Randomized Interventional Trials (SPIRIT) (*Chan et al., 2013*), the CONSORT guidelines (*Schulz, Altman & Moher, 2010*) (Fig. 1), and the Template for Intervention Description and Replication (TIDieR) Checklist (*Hoffmann et al., 2014*).

The study population will be people diagnosed with low back pain for ≥ 3 months. Participants will be recruited from the clinical center of the Santo Tomás University, through social media and publications health centers in the city of Talca.

#### Ethics

The study was approved by the Central-South Macrozone Ethics Committee of the Universidad Santo Tomás, Chile, according to the Declaration of Helsinki for biomedical research (exp-23-13). All participants will provide written informed consent.

### Eligibility criteria

Inclusion criteria will be men and women aged 45–60 years (*Knauer, Freburger & Carey, 2010*), non-specific low back pain ≥ 3 months without compromise of any lower limb, average pain intensity ≥ 3/10 and ≤8/10 (according to the 0–10 numerical rating scale (NRS)) in the last month. Exclusion criteria will be psychiatric, neurological or oncological

 

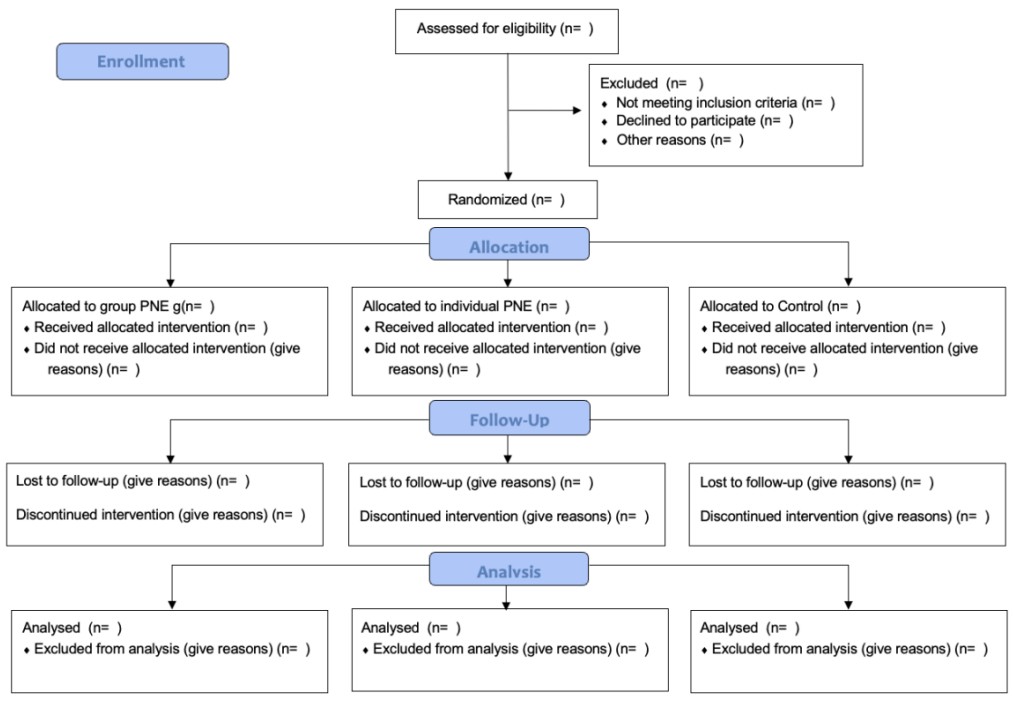

**Figure 1** The proposed CONSORT diagram of enrollment, allocation, follow-up, and analysis throughout the study for each arm.

diseases, operated of some lumbar pathology, chronic low back pain due to a specific cause (lumbar stenosis, herniated disc, spinal deformity, fracture, spondylosis), be receiving some form of active or passive physical therapy for pain at the time of the investigation or having received it in the last two months, and have previous experiences with PNE.

## Intervention

The content of the intervention will be identical in the experimental groups. The only difference is that the PNE in the in-group modality (G1) is provided in groups of 3–5 participants, while in the individual modality (G2) it is provided directly to a single participant. The intervention has an active educational approach based on reconceptualizing the maladaptive beliefs that influence the fear-avoidance behavior of the participants through updated contents of the neuroscience of pain. This will be delivered in a single face-to-face session of approximately 60–80 min in the kinesic clinic of the Santo Tomás University and will be delivered through active participation encouraging synchronous discussion of the information and allowing note-taking. A break will be made in the middle of the session to allow the participants a break. Five key domains will be structured based on the Fear and Belief Avoidance Questionnaire (Table 1) which will serve as a guide for the sessions through a Powerpoint presentation. In addition, participants will be encouraged to stay active by walking for 20–30 min, 3–5 times a week and will be taught an exercise to improve the activation of the transverse abdominis (bracing or abdominal following). The PNE will be delivered by a physical therapist with four years of experience

**Table 1  Summary content of pain neuroscience education.**

| Dimension | Content |
|---|---|
| Dimension 1: Approach | Introduction to the program. |
| | The objective of today's session will be to contrast what we know about pain, with the scientific advances in rehabilitation through a conversation. |
| | Your opinion regarding each of the points that we will discuss is very important for your rehabilitation. |
| Dimension 2: Pain as protector (the big picture) | Pain as a multidimensional experience |
| | - Neurophysiology of pain: pain pathways, neuronal synapses, action potentials and pain perception. Gate of entry of pain. |
| | - Modulation of pain: explain the processes of facilitation and inhibition. The theory of gate control will be emphasized. |
| | - Acute pain and chronic pain: explain the differences between acute pain and chronic pain |
| Dimension 3: Pain is not always an indicator of damage (pain $\neq$ damage) | Pain does not depend entirely on the state of the tissues. |
| | Imaging findings associated with aging are like wrinkles or gray hair. |
| | examples of pain in the absence of damage (eg headache) |
| Dimension 4: Movement as an opportunity for recovery (movement experience) | -Importance of movement |
| | -explain the relevance of movement from the plasticity and robustness of the human body. |
| | -Neuroplasticity of the nervous system: Explain neuroplastic changes due to experience and learning with an emphasis on positive changes through movement. |
| | benefits of movement in functionality |
| | progressive active strategies |
| Dimension 5: Recovery expectations | Concept of pain modulation from cognitions |
| | strategies to help manage pain, relaxation/breathing techniques, positive thinking |

in this intervention. In addition, the same physical therapist will provide the therapy to both experimental groups to maintain the same patient-physical therapist relationship and not to influence the results due to this contextual aspect, so no adaptations were made for each group either.

In addition, a brochure will be delivered with the main points of each of the domains and information capsules will be made to which the participants will have access (five 15-minute videos, one per domain). Participants will be instructed to record on a calendar the days they performed the walks, exercises, and read the brochure, and/or review the information capsules to assess compliance with the treatment and for each domain invent a metaphor or write how they would explain it to another person. This activity must be delivered in the second assessment.

The total time of the intervention will be 135–155 min for both groups.

## Outcomes measures
### Demographic information
Patients will self-report a variety of demographic and descriptive information *via* paper forms, including age, sex, comorbidities, medications commonly used, duration of

symptoms, marital status, employed status, educational level, and economic income. Weight and height will be measured at the beginning of the assessments.

The social determinants will be categorized as follows: (I) employment status: employed *versus* unemployed; (II) educational level: participants will assigned to the lower educational level if they had not completed secondary education and to the higher educational level if they had completed secondary education or university studies (*Núñez Cortés et al., 2023a*; *Nuñez Cortés et al., 2023b*); (III) economic income will be categorized according to individual monthly taxable income. The cut-off point will be set at a value equal to or less than USD 545, which determines the degree of coverage provided by the Chilean public health system.

## Primary outcomes
### Fear avoidance and beliefs
This variable will be evaluated with the Fear Avoidance Beliefs Questionnaire (FABQ). The FABQ presents a minimum detectable change of 5.4 for the physical activity subscale and 6.8 for the work subscale (*George, Valencia & Beneciuk, 2010*). The minimal clinically important change (MCID) is 4 points for the FABQ-Physical Activity scale (FABQ-PA) and 7 points for the FABQ-Work scale (FABQ-W) (*Monticone et al., 2020*). Its Spanish version has been validated with a high internal consistency (Cronbach $\alpha = 0.933$) with good test-retest reliability (ICC = 0.966) (*Kovacs et al., 2006*). Higher scores indicate higher levels of fear avoidance beliefs (*George, Valencia & Beneciuk, 2010*).

## Secondary outcomes
### Algometry
An algometer will be used to measure pressure pain sensitivity. The average of three measurements will be used (*Nussbaum & Downes, 1998*; *Christidis, Kopp & Ernberg, 2005*). This method has excellent test-retest reliability (ICC = 0.80–0.99) within a session and between sessions (ICC = 0.87–0.95) (*Potter, McCarthy & Oldham, 2006*). It also presents good to excellent inter-evaluator reliability (ICC = 0.74–0.89) (*Nussbaum & Downes, 1998*). The minimum significant change has been reported at $\geq 1.16$ kg/cm$^2$/s (*Fuentes et al., 2011*). Patients will be evaluated lying down in a comfortable position according to the area to be evaluated with a digital algometer (WAGNER FDX10). A gradual increase in pressure of 1 kg/cm$^2$/s will be applied bilaterally at five cm lateral to the spinous process of L3, in the second metacarpal and the tibialis anterior muscles with the aim of examining changes in generalized sensitivity to pressure pain (*Roussel et al., 2013*). Between each repetition there will be a rest of approximately 30 s.

### Pain self-efficacy
The pain self-efficacy questionnaire (PSEQ) will be used to assess this variable (*Koenig et al., 2014*). It has a high internal consistency (Cronbach $\alpha = 0.92$) (*Nicholas, 2007*), high test-retest reliability (ICC = 0.86) and a minimal clinically important difference between 5.5 and 8.5 points (*Dubé, Langevin & Roy, 2021*). Higher scores indicate stronger self-efficacy beliefs, while low scores indicate a subject more focused on their pain.

### Catastrophizing

Catastrophizing will be evaluated by applying the Pain Catastrophizing Scale (PCS) (*Burri et al., 2018*). In a population with chronic pain, has a high total internal consistency (Cronbach $\alpha = 0.92$), a moderate total test-retest reliability (ICC = 0.73) (*Lamé et al., 2008*), and is validated in Spanish (*García Campayo et al., 2008*).

### Pain intensity

Pain intensity will be measured with the Numerical Rating Scale (NRS) since it presents minimal translation difficulties, which allows its use in all cultures and languages (*Karcioglu et al., 2018*). The NRS presents, in patients with chronic lumbar pain, an excellent test-retest reliability (ICC = 0.92), a standard error of measurement of 0.86, a minimum detectable change of 2.4 points, and a clinically important minimum change of 4 points (*Maughan & Lewis, 2010*). The intensity at rest and activity in the last 7 days will be considered.

### Treatment expectation

The treatment expectation questionnaire (TEX-Q) will be used. This is a generic multidimensional measure that allows evaluating the patient's expectations in both medical and psychological treatments and allows comparing the impact of multidimensional expectations in different conditions. Its psychometric properties have yet to be determined, but it was developed through a rigorous procedure that incorporated complex and diverse literature and expectation evidence as well as peer review (*Alberts et al., 2020*).

### Sample size

Sample size calculation was performed with G*Power 3.1. A repeated measures analysis of variance (ANOVA) model with within-between interaction, was used. Assuming an alpha risk of 0.05, a power of 0.95, a correlation between repeated measures of 0.5, a 10% drop-out rate, and a small effect size (0.26), a total of 20 participants per group (three groups) were required. The effect size estimate was based on a previous study about education compared to physical therapy for the FABQ outcome (*Marshall et al., 2022*) and the recommended effect size for clinical studies (*Lakens, 2013*).

### Randomization and blinding

Participants will be randomized in a 1:1:1 ratio between intervention and control arms using balanced group assignment with block randomization with permuted block size. An independent researcher will carry out the process of randomization through a web platform (http://www.randomizer.org) and will allocate concealment from patients and other investigators using sealed, opaque envelopes.

Participants, assessor, and statistician will be blinded to group assignment. Participants will only be informed that they will receive an educational intervention without indicating to which experimental group they will belong; the assessor will not be informed of the group to which the participants belong; and the identification data of the participants will be coded in the database before being sent to the statistician. However, the physical therapist who will perform the educational intervention will not be blinded.
### Data collection and management

Outcome measures will be collected at one-week before intervention, one-week post-intervention and four-weeks post-intervention (Fig. 2).

## Data analysis
### Statistical approach

The analysis of the data will be blinded, as each subject and condition will be coded by a consultant who will not be involved in the investigation. SPSS software version 25.0 will be used for all analyzes (SPSS Inc. Armonk, NY, USA). Data will be reported as mean with SD or median with interquartile range. Statistical analyzes were performed on an intention-to-treat basis, with imputation of missing data using the average of the remaining participants.

Normality tests will be performed using the Shapiro–Wilk test given the number of subjects needed for each group. In addition, the homogeneity of variance and the sphericity of the data will be determined. In the case of not assuming any of these assumptions, the Green-Hausser correction will be used for the interpretation of subsequent analyzes. A three-way repeated measures ANOVA will be conducted to analyze both primary and secondary outcomes. The factors considered in this analysis will include time and intervention. Additionally, the third factor will encompass variables such as employment status, education level, or income. In case of significant interaction, a two-way repeated measures ANOVA (time (pre-post) × intervention (in-group PNE, individually PNE, no intervention)) will be performed for each employment status (employed or unemployed). Furthermore, a two-way repeated measures ANOVA (time (pre-post) × intervention (in-group PNE, individually PNE, no intervention)) will be performed for each educational level (high or low). Lastly, a two-way repeated measures ANOVA (time (pre-post) × intervention (in-group PNE, individually PNE, no intervention)) will be performed for income (≤USD 545, or >USD 545). If interactions are detected in any of the 2-way ANOVAs, a *post-hoc* analysis will be conducted using multiple pairwise comparisons employing a *t*-test corrected by Bonferroni. The significance level will be set at 5%.

A stepwise multiple linear regression will be performed to estimate the influence of changes in primary and secondary outcomes. In each model, adjustments will be made for employment status, educational level, and income. In this analysis, the aforementioned variables will be sequentially incorporated into each model.

The calculation of Cohen's d will be performed to determine the effect size (ES) of all intragroup variables. Consider a small effect size if it is less than or equal to 0.2; medium of 0.3 to 0.5 and large of 0.5 to 0.8 (*Jacob, 1992*).

## DISCUSSION

CLBP is considered to be one of the most prevalent health conditions, contributing significantly to the global burden of disease (*Rabiei, Sheikhi & Letafatkar, 2021*). Therefore, it is relevant to determine cost-effective therapeutic strategies that can improve the clinical condition of patients. In this sense, therapeutic strategies directed by physical

| | STUDY PERIOD | | | | | | |
|---|---|---|---|---|---|---|---|
| | Enrolment | Allocation | Post-Allocation | | | | Close-out |
| Timepoint | -t1 (pre-baseline) | 0 (pre-baseline) | t1 (week 0) | t2 (week 1) | t3 (week 2) | t4 (week 5) | tx (post study) |
| **ENROLMENT** | | | | | | | |
| Eligibility screen | x | | | | | | |
| informed consent | x | | | | | | |
| Demographic characteristic | x | | | | | | |
| Allocation | | x | | | | | |
| **INTERVENTIONS** | | | | | | | |
| PNE interventions (in-goupr and individually) | | | | x | | | |
| Control (usual care) | | | | | | | |
| **ASSESMENTS:** | | | | | | | |
| Fear Avoidance and Beliefs (FABQ) | | | x | | x | x | |
| Sensitivity to pressure pain (Algometry) | | | x | | x | x | |
| Pain Self-efficacy (PSEQ) | | | x | | x | x | |
| Catastrophizing (PCS) | | | x | | x | x | |
| Pain Intensity (PI-NRS) | | | x | | x | x | |
| Treatment expectation (TEX-Q) | | | x | | x | x | |
| Statistical Analysis | | | | | | | x |

**Figure 2  Flow diagram of the planned protocol pathway.**

therapists based on the biopsychosocial (BPS) model have been shown to effectively improve symptoms in people with spinal disorders (*Miki et al., 2023*).

PNE is based on the BPS model (*Nijs et al., 2011*; *Moseley & Butler, 2015*) and has been shown a to have positive effect on pain, disability, catastrophism, kinesiophobia, physical performance, and a reduction in health care costs in subjects operated for radiculopathy (*Louw et al., 2011*; *Louw et al., 2016a*; *Gallagher, McAuley & Moseley, 2013*; *Mittinty et al., 2018*). This may be because it influences pain cognitions, an important aspect in the vicious circle of central sensitization in patients with CLBP even when not all present with such sensitization (*Huysmans et al., 2018*). However, PNE implemented in isolation may not generate significant clinical effects (*Louw et al., 2016c*; *Louw, Puentedura & Zimney, 2016*; *Puentedura & Flynn, 2016*). While PNE performed in a group modality could potentially facilitate learning through social observation, which would provide a positive influence on therapy due to the observation of the behaviors exhibited by other participants (*Romm et al., 2021*).

The lack of effectiveness of the PNE applied in isolation may be because it has been approached mainly from the neurophysiology of chronic pain with little orientation of the contents to the context of the person. Furthermore, a very small number of studies have considered within the demographic characteristics the educational level of the subjects (*Malfliet et al., 2017*; *Mittinty et al., 2018*; *Rufa, Beissner & Dolphin, 2018*); and of these, only one used this data to be analyzed, however, it was used as a secondary variable and it was evidenced that a high educational level is associated with the expectation of recovery, not with the very effectiveness of the intervention of education (*Mittinty et al., 2018*), In addition, these studies tended to present more subjects with high educational levels, however, it has been shown that people who are more predisposed to have lumbar pain chronification have a low level of education (*Meucci et al., 2013*).

It follows that the studies have not considered in their analysis the possible differences in the effectiveness that may exist in the application of pain neuroscience education with orientation to the context according to the educational level and other SDH related to the socioeconomic level of the subjects to whom this intervention is applied, evidencing a knowledge gap since this could influence the effects of this type of educational therapy.

This study will provide new data on the efficacy of pain neuroscience education focused on fear-avoidance beliefs on clinical and psychosocial variables in patients with CLBP, differentiating the effects between in-group and individual approaches and the influences of social determinants of health. This will allow the identification of strategies for the implementation of PNE in clinical contexts that allow a better cost-effectiveness of the intervention.

## LIMITATIONS

One of the main limitations of the study is that the intervention will only be carried out in one session, so if doubts arise among the participants after the face-to-face session, they cannot be resolved by the physical therapist, and this could harm the interpretation and acquisition of information. In addition, since PNE is little known by the general population,

the expectation of the effectiveness of the therapy can significantly influence the results. Finally, the physical therapist who will apply the PNE will not be blinded. However, both experimental groups will be instructed to deliver the content in the same way.

### Funding
The authors received no funding for this work.

### Competing Interests
Guillermo Mendez-Rebolledo is an Academic Editor for PeerJ.

### Author Contributions
- Joaquín Salazar-Méndez conceived and designed the experiments, performed the experiments, prepared figures and/or tables, authored or reviewed drafts of the article, registration materials, and approved the final draft.
- Iván Cuyul-Vásquez conceived and designed the experiments, performed the experiments, prepared figures and/or tables, authored or reviewed drafts of the article, and approved the final draft.
- Felipe Ponce-Fuentes conceived and designed the experiments, performed the experiments, prepared figures and/or tables, authored or reviewed drafts of the article, and approved the final draft.
- Rodrigo Núñez-Cortés conceived and designed the experiments, performed the experiments, authored or reviewed drafts of the article, and approved the final draft.
- Guillermo Mendez-Rebolledo conceived and designed the experiments, performed the experiments, analyzed the data, authored or reviewed drafts of the article, and approved the final draft.
- Jorge Fuentes conceived and designed the experiments, performed the experiments, prepared figures and/or tables, authored or reviewed drafts of the article, and approved the final draft.

### Clinical Trial Ethics
The following information was supplied relating to ethical approvals (*i.e.*, approving body and any reference numbers):

The study was approved by the Central-South Macrozone Ethics Committee of the Universidad Santo Tomás, Chile, according to the Declaration of Helsinki for biomedical research (exp-23-13).

### Data Availability
This is a registered report.

### Clinical Trial Registration
The following information was supplied regarding Clinical Trial registration:

NCT05953454.

## Supplemental Information

Supplemental information for this article can be found online at http://dx.doi.org/10.7717/peerj.17507#supplemental-information.

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
