# Peer review of "Effectiveness of in-group versus individually administered pain neuroscience education on clinical and psychosocial outcomes in patients with chronic low back pain: randomized controlled study protocol"

_PeerJ, doi:10.7717/peerj.17507_

## Round 0.1 · original submission · Major Revisions

Please revise the manuscript as per the comments of both peer reviewers.

**Language Note:** The review process has identified that the English language must be improved. PeerJ can provide language editing services - please contact us at [email protected] for pricing (be sure to provide your manuscript number and title). Alternatively, you should make your own arrangements to improve the language quality and provide details in your response letter. – PeerJ Staff

Reviewer 1 ·

Basic reporting

First of all, congratulations to the authors for their research. Investigating the influence of social determinants on pathologies of such high incidence as CLBP is of high clinical relevance.

Line 28: the authors states "Sixty-nine participants with CLBP"; nevertheless, echa group included 33 participants, which implies a total of 99 participants. Please correct

Line 78: "...there are no other studies that analyzing..." did you mean "that analize..."


The introduction makes clear the importance of social determinants in health and neuroscience-based education on low back pain and associated comorbidities, however, it lacks solid support for the hypothesis. The authors hypothesize "that there will be significant differences in favor of the group intervention in-group over the individually intervention groups", nevertheless, this is only supported by Romm et al. (2021) citation. I understand that the authors assume that social learning in therapeutic interaction is one of the factors that would justify these differences, however, the structure of the paragraph does not make the idea completely clear. This idea could be deepened, in order to strengthen the hypothesis raised.

Line 77 - 81: Tha authors states "On the other hand, one study suggests that the educational level does not influence the effectiveness of the PNE (Bilterys et al., 2022)". This idea seems isolated from the rest of the paragraph. It does not refute or complement what was stated above regarding in-group therapy. Nor is the controversy regarding educational level or other social determinants of health discussed. If the statement aims to justify the second hypothesis, the idea must be developed further. The relevance of social determinants of health on CLBP symptoms has been previously explained, however, it is possible that these have not been addressed in the PNE. The lack of effectiveness of certain PNE modalities, or the differences observed between them, could be due to the non-consideration of HSD as the educational level. Although it is assumed that it does not influence the effectiveness of PNE, there is a lack of clarity regarding other HSD or studies that compare different modalities of PNE in consideration of these factors.

Some of these arguments have been presented in the discussion of the protocol. Consider including in the introduction

Line 141 - 146: change tense. i.e: was --> will

Experimental design

LIne 98-100: Please speciffy how the numbers of participantes will be determined. Also specify how patinets will be assigned to each group

Line 112-113: In the exclusion criteria the authors states "have recieved any modality of active or pasive physical therapy..." Since evaluations will be performed one week before, one and four weeks post intervention, consideration should be given to excluding subjects who are currently enrolled in a physical therapy program.

Line 123-124: The authors states that participants will be encouraged to walk for 20-30 min 3 at 5 times a week. How will it be controled? There should be a mechanism which allows to at least partially control the physical activity level.

Validity of the findings

No comment

Additional comments

The protocol is well structured, easy to understand and within the current interest of the discipline. It is necessary to reinforce the argumentation of the hypothesis, although the restructuring of the discussion could fulfill this point.

Reviewer 2 ·

Basic reporting

- In the intervention section (line 115), there's a lack of clear differentiation between the interventions implemented for the in-group and individual experiment groups. Enhancing this section with more detailed explanations on how these two groups were treated differently during the execution of the experiment would greatly improve clarity and understanding.

- In line 199, replace “a loss of 10%” with 10% drop-out rate or loss to follow-up rate

Experimental design

- The experiment includes two primary outcomes: Fear avoidance and beliefs + Algometry. How is sample size determined by multiple primary outcomes given that the effect size estimation may be totally different in these two outcomes?

- In line 199, the sample size calculation methods used specific parameters, including a correlation of 0.5 between repeated measures and a small effect size (f) of 0.26. How are these values determined respectively? While the authors briefly mentioned the determination of the effect size using a cited formula, further elaboration on the meaning of parameters and their intuitive interpretation would enhance comprehension.

Validity of the findings

- In line 234, Bonferroni correction was used for multiple comparison adjustment when there exists significant interaction between factors. How many interaction terms are used in the model, intervention*time only or intervention*other factors (employment status, educational level, and income) as well? Is Bonferroni correction also applied to account for multiple comparison issues for multiple outcomes?

- In line 237, the authors can elaborate more on how stepwise regression is performed. FOr example, what are the criteria for model selection at each step? Also, since this is a statistical inference question, It would be recommended to use “estimate the influence of …” instead of “predict the influence of …”

---

## Round 0.2 · accepted · Accept

Thank you for the revised manuscript which has been accepted.

Reviewer 2 ·

Basic reporting

I think that the authors have adequately addressed the comments made by the reviewers in the revised version of the manuscript. Therefore, I have no further comments.

Experimental design

I think that the authors have adequately addressed the comments made by the reviewers in the revised version of the manuscript. Therefore, I have no further comments.

Validity of the findings

I think that the authors have adequately addressed the comments made by the reviewers in the revised version of the manuscript. Therefore, I have no further comments.